# Analysis of the Barriers and Motives for Practicing Physical Activity and Sport for People with a Disability: Differences According to Gender and Type of Disability

**DOI:** 10.3390/ijerph20021320

**Published:** 2023-01-11

**Authors:** Josu Ascondo, Andrea Martín-López, Aitor Iturricastillo, Cristina Granados, Iker Garate, Estibaliz Romaratezabala, Inmaculada Martínez-Aldama, Sheila Romero, Javier Yanci

**Affiliations:** 1Physical Education and Sport Department, Faculty of Education and Sport, University of the Basque Country, UPV/EHU, 01007 Vitoria-Gasteiz, Spain; 2Faculty of Education and Sport, University of the Basque Country, UPV/EHU, 01007 Vitoria-Gasteiz, Spain; 3Society, Sports and Physical Exercise Research Group (GIKAFIT), Physical Education and Sport Department, Faculty of Education and Sport, University of the Basque Country, UPV/EHU, 01007 Vitoria-Gasteiz, Spain; 4Disability Research Department, Fundación GaituzSport Fundazioa, 48003 Bilbao, Spain

**Keywords:** motor practice, motivation, habits, impediments, disability

## Abstract

The aim of the study was to analyse the perception of the barriers and motives for the practice of physical activity (PA) in people with a disability, according to gender and type of disability. The participants in this investigation were 103 people with a disability (33.25 ± 11.86 years) who were habitual users of PA or sports programmes. They completed the questionnaire Motives and Barriers for Physical Activity and Sport (MBAFD). The results showed that personal barriers were more important than environmental ones and that the most outstanding motives were those related to leisure, enjoyment or social aspects. Regarding gender, women perceived more barriers than men. For the people with physical (PD) and intellectual (ID) disabilities, the most common barriers were of an intrinsic nature, and for those with a visual disability (VD), barriers of an environmental nature. With respect to the motives, people with PD gave higher scores to the items related to aspects of physical improvement and rehabilitation. In contrast, people with ID and VD placed more importance on reasons of leisure, enjoyment and social relations. Knowledge of these findings can be a tool to help increase the provision of PA for people with a disability.

## 1. Introduction

Worldwide there are about 650 million people with a disability, which corresponds to about 10% of the world population [1]. Specifically in Spain, it is estimated that about 4.38 million people have some type of disability, of whom 58.7% are women and 41.3% are men [2]. Despite the fact that people with a disability have the right to participate in society under equal conditions as the rest of us [3], it has been reported that this population has greater difficulties for participating in society, due to the fact that they suffer a high level of social exclusion which affects fields including education or participation in different social activities [4]. This situation means that people with a disability are subject to the stereotypes, stigmas and prejudices established by society. It has also been reported that this group more often faces different barriers regarding social and/or political communication and accessibility, among others, which affect their general wellbeing and prevent them from having equal opportunities [5].

One of the strategies for improving general wellbeing and encouraging equal opportunities in people with a disability is the practice of physical activity (PA) [6,7]. It has been stated that the practice of PA can help them gain greater inclusion in society [8,9]. Some previous investigations have observed that the practice of PA brings them psychological benefits, improving self-esteem, autonomy, goal achievement, personal development, self-control and self-confidence [9,10]. Furthermore, it has been shown that the practice of PA has social benefits as it favours inclusion and social relation [9,10]. As well as the psychological and social benefits, some researchers [11,12] state that the practice of PA by people with a disability also provokes physical benefits such as better fitness, the ability to execute movements and functionality permitting greater autonomy and self-sufficiency. Bearing in mind that it has been said that the practice of PA benefits this population both in their physical and mental health [13], it is necessary for them to have adequate levels of PA practice. However, there are many studies that show that people with a disability have a lower level of PA than people without a disability [14,15,16,17] revealing a high level of sedentarism [18,19,20]. In this line of research, it has been reported that people who suffer some types of disability are between 16 and 62% less likely to comply with the recommended levels of PA and present a greater risk of developing health complications as a result of physical inactivity [21]. Similarly, Ramírez et al. [22] in a study of an adult population with disability observed that only 29% of the participants carried out more than the 60 min daily recommended by the World Health Organisation [23] and that 51% of the women and 40.7% of the men were categorised as sedentary.

In order to favour the practice of PA in people with a disability, it has been stated that it is necessary, on the one hand, to know the motives and, on the other, the barriers that are perceived regarding the practice of PA [7,24]. In this way, action can be taken to adjust the provision of PA to their motivations and take action to minimise the barriers they perceive. Possibly, due to the importance of these results, previous studies have analysed the motives for and barriers to the practice of PA both in people with a disability who are not habitual practitioners of PA or sport [25,26,27] and in sportspeople or habitual practitioners of PA with a disability in different modalities, at different competitive levels and with different impairments [28,29,30,31,32,33]. Moreover, it has been shown that there are significant differences in the perception of barriers between different disabilities. On the one hand, Juanbeltz et al. [32] found that people with intellectual disability (ID) show a greater refusal to engage in PA because they do not like it compared to people with Down Syndrome. On the other hand, people with physical disabilities (PD) give more weight to not feeling able to make time for PA compared to people with ID. However, most studies have been carried out with groups with a specific disability, with limited samples or without taking types of disability or gender of the participants into consideration. The majority of the studies expound the need to carry out an analysis of the motives and barriers in larger samples and study the differences according to the type of disability and gender.

Therefore, the objectives of the present study were, on the one hand, to discover the perception of the barriers to and motives for the practice of PA and sport of people with a disability and, on the other, to analyse the differences which exist in the perception of the motives and barriers according to gender and the type of disability. The hypotheses of this study were that women perceive more barriers to practice PA than men, people with ID and/or PD perceive intrinsic barriers to a greater extent than environmental barriers and people with VD perceive environmental barriers to a greater extent than intrinsic barriers

## 2. Materials and Methods

### 2.1. Participants

The participants in this study were 103 people with a disability between 15 and 57 years of age (33.25 ± 11.86 years), of whom 40 were women (36.85 ± 11.64 years) and 63 were men (30.96 ± 11.51 years). All the participants were habitual users of some type of physical exercise or sports programme carried out in the Basque country (Spain), with a mean of 14.38 ± 10.40 years’ experience in such activities. Of the total number of participants, 29.1% had a physical disability (PD, *n* = 30, 35.46 ± 11.19 years), 34.0% an intellectual disability (ID, *n* = 35, 29.85 ± 12.64 years), 34.0% a visual disability (VD, *n* = 35, 33.91 ± 11.41 years) and 2.9% did not respond to the question of what type of disability they had (*n* = 3, 43.0 ± 2.64 years). The inclusion criteria were that the subjects had been carrying out PA and sport at least for the previous two years and that they had a disability certificate. The project was approved by the Ethics Committee for Research with Humans Beings (CEISH, code M10_2019_058) from the University of the Basque Country (UPV/EHU) and followed the requirements established in the Declaration of Helsinki (2013).

### 2.2. Procedures

The questionnaire Motives for and Barriers to the Practice of Physical Activity and Sport *(MBAFD)* was administered to all the participants in the study in order to know their perception of the barriers and motives for practising PA and sport. The questionnaire was filled in by each participant individually, on paper and in the presence of the researcher. In the case that the participant so requested, the researchers offered clarifications to each of the questions raised in the questionnaire. Before completing the questionnaire, all the participants received detailed information of the procedure for its adequate completion.

### 2.3. Measurements

The questionnaire on Motives for and Barriers to the Practice of Physical Activity and Sport MBAFD was used to discover the barriers and motives perceived by the participants when they performed PA and sport. It was previously validated by Kostiuk [34] for the adult population and has been used by several researchers in people with disabilities who practice PA and sport [29,32,33]. Kostiuk [34], in a validation study, analysed a small sample of people with a disability, reiterating the suitability of the instrument for this specific group. The questionnaire used consists of 53 items or questions divided into two blocks: the first block (27 items) refers to the barriers to the practise of PA and sport, and the second block (26 items) deals with the motives for practising PA and sport. All the questions are answered using a Likert-type scale (1–4) where 1 is not at all and 4 a lot; it includes, also, the option of don’t know/no opinion (DK/NO).

### 2.4. Statistical Analsysis

The descriptive results are presented as frequencies and percentages of the replies given by the participants in each item or question. Cronbach’s Alpha statistic was used to describe the internal consistency of the questionnaire. The chi squared statistic (Chi^2^) was used to analyse the differences according to gender and type of disability in the distribution of the replies. The analyses were performed with the Statistical Package for Social Sciences (SPSS Inc, version 26.0, Chicago, IL, USA). Statistical significance was established at *p* < 0.05.

## 3. Results

The internal consistency (Cronbach’s Alpha) of the total items (*n* = 53 items) was 0.84. Regarding between-items’ internal consistency of the different blocks, Cronbach’s alpha values were as follows: 0.86 for the first block (barriers to the practise of PA and sport, *n* = 27 items) and 0.83 for the second block (motives for practising PA and sport, *n* = 26 items).

Table 1 presents the results of the barriers perceived by all the participants for the practice of PA and sport. The results obtained show that the most important items with regard to the barriers were the following: “I have a disability”, “I don’t feel happy with my body”, “I don’t feel like it”, “I am not able to make time for myself”, “My health is not good” and “The technical staff are not suitable”. In contrast, the items “I don’t think it is necessary to do exercise”, “I don’t like it”, “I am too old”, “I am embarrassed to be seen doing it” and “There is no provision for the activity I like” presented the lowest values.

Table 2 shows the results of the motives for practising PA and sport perceived by the total sample of participants. The highest values were obtained with the items “Because I like doing this physical activity”, “Because of the positive sensations that doing physical activity gives me”, “Because I feel good when I do physical activity”, “For enjoyment” and “To make contact with other people“. In contrast, the items “To share activities with my children”, “To spend more time with my partner”, “To recover from an illness or injury” and “Because of my doctor’s advice” showed the lowest values.

Table 3 shows the results referring to the barriers to the practice of PA and sport according to gender. The group of women scored higher values (*p* < 0.05) than the group of men in the items “Fear of hurting myself”, “My health is not good”, “The technical staff are not suitable”, “I don’t feel like it”, “I am not fit”, “I have no one to do exercise with” and “I don’t like it”. However, the men’s group only had higher scores (*p* < 0.05) in one single barrier (“I am too old”). It should be noted that gender differences would change when the type of disability is taken into account.

Table 4 presents the results on the motives for practising physical activity according to gender. The women’s group obtained significantly higher values (*p* < 0.05) in the motives “For the positive sensations that doing physical activity gives me”, “Because I want it to be part of my life style”, “To recover from an illness or injury”, “Because of my doctor’s advice” and “To spend more time with my partner”. In contrast, the men’s group recorded higher values (*p* < 0.05) in the items “For enjoyment”, “Because I like to overcome challenges”, “To disconnect for a while” and “To eliminate stress”.

Table 5 shows the results referring to the differences in the perceived barriers to the practice of PA and sport according to the type of disability. People with PD recorded higher values (*p* < 0.05) than people with VD and ID in the items “I can’t make time for myself“, “My health is not good”, “I don’t feel happy with my body” and “I have to look after my children”. In contrast, the items “I don’t feel like it”, “I’m tired” and “I have to look after older relatives” obtained higher scores (*p* < 0.05) in the group of people with ID. The group of participants with VD recorded higher scores (*p* < 0.05) in the item “There are no spaces nearby where I can practise”.

Table 6 presents the results referring to the differences in the perceived motives for practising PA and sport according to the type of disability. The items “To control my weight”, “To recover from an illness or injury”, “To improve my appearance”, “Because of my doctor’s advice” and “To share activities with my children” obtained a higher score (*p* < 0.05) in people with PD than those with ID or VD. However, the items “To make contact with other people” and “Because it allows me to be in natural surroundings” had higher scores (*p* < 0.05) in people with ID. For their part, the people with VD recorded higher scores (*p* < 0.05) in the items “Because I like doing this physical activity”, “For enjoyment”, “Because I like to overcome challenges”, “Because I want it to be part of my life style”, “To improve my mental wellbeing” and “Because I feel bad if I don’t practise”.

## 4. Discussion

The main objective of this study was to discover the perception of barriers to and motives for the practice of PA and sport in people with a disability, also analysing the existing differences depending on gender and type of disability. Most studies on this topic have not taken into account the gender and type of disability of the participants [28,29,31,33]. This is why the contribution of this paper is the analysis of the aforementioned motives and barriers in a wider sample than that habitually used and the distinguishing among the type of disability and the gender of the participants. The results obtained in this study show that personal barriers are more important than environmental barriers and that the motives that received the highest scores were those related to leisure, enjoyment and social aspects. Regarding the gender of the participants, the women perceived more barriers than the men to the practice of PA and sport. With respect to the type of disability, we found differences both in the barriers and in the motives for practising PA and sport. Intrinsic barriers predominate in people with PD and ID, and environmental barriers predominate in people with VD. Among the motives for practising PA and sport, the people with ID and VD gave higher scores to motives of leisure, enjoyment and social relations, while the people with PD focussed more on physical improvement and rehabilitation. These findings can be of great use in knowing which are the existing motives and barriers regarding the practice of PA and sport in people with a disability and, thus, to be able to offer different physical-sporting activities that are attractive for people with a disability and be able to increase the levels of practice.

Knowing the motives for and barriers to the practice of PA and sport perceived by people with a disability can provide information for acting according to their needs and for being able to increase the levels of PA and sport practice [30]. The findings of the present study show that the most important perceived barriers were those of an intrinsic/personal nature—including, for example, “I have a disability”, “I don’t feel happy with my body”, “I don’t feel like it”, among others—while there is only one environmental barrier with a high score: “The technical staff are not suitable”. These results coincide with previous studies where it was observed that the most important barriers were of an intrinsic nature. For example, Juanbeltz et al. [32], in a study carried out with active people with PD, also observed that the most important barriers were personal (“I have a disability”, “My health is not good”, “I don’t feel like it”, etc.). Similarly, Mendia et al. [33], in a study carried out with athletes with ID and cerebral palsy (CP) that competed in the Spanish soccer Liga Genuine, reported that the barriers that obtained the highest scores were those of a personal character, such as “I don’t feel happy with my body“ and “I don’t feel like it”. In contrast to the findings of this study, Abellán and Januário [28] in athletes with ID; Avalos et al. [29] in athletes with VD practising goalball; and Jaarsma, Gertzen, et al. [30] in Dutch Paralympic athletes with PD, both ambulant and using wheelchairs, reported that the major barriers were of an environmental nature. Possibly these differences in the perception of the personal or environmental barriers can be due to how the disability affects the individuals. People with a disability that have a more limited access to the practice of PA and sport may perceive greater environmental barriers, but people with less limited accessibility may perceive personal barriers to a greater extent.

With respect to the motives for practising PA and sport, the results obtained in the present study indicate that the main motives for practicing PA and sport show a strong play-, recreation- and social-oriented nature: “Because I like doing this physical activity”, “For the positive sensations that doing physical activity gives me”, “Because I feel good when I do physical activity“, “For enjoyment”, “To make contact with other people”. In contrast, the less important motives for practising were “To share activities with my children”, “To spend more time with my partner” and “Because of my doctor’s advice or for rehabilitation”. The results obtained referring to the motives in our study coincide with the studies by Avalos et al. [29] carried out with goalball players with VD, by Jaarsma, Geertzen et al. [30] with Dutch Paralympic athletes, by Juanbeltz et al. [32] with physically active athletes with a disability and by Mendia et al. [33] with football players from the Liga Genuine with ID and CP, in which it was also observed that the main motives for practising PA and sport are related to wellbeing, enjoyment and play aspects. Abellán and Janúario [28] in a study of people with ID and Úbeda-Colomer et al. [27] in a study with Spanish university students with different types of disability indicate that the motives related to improving health are also very important. In the case of Abellán and Janúario [28] the health-related motives are especially important because people with ID have greater vulnerability to health problems, particularly those related to sedentarism. In the study by Úbeda-Colomer et al. [27], 70.5% of the sample had a recognised disability percentage between 33% and 64%, and the rest were more seriously affected. Furthermore, the participants had a mean age of 34.66 ± 12.02 years, which indicates that it was a sample of people at an age at which health may gain greater relevance. Possibly, in groups whose health is more at risk, the motives given for practising PA and sport are more in the line with improving their health; however, in groups with fewer health risks, the motives for practising PA and sport are more oriented to play, recreation and social aspects. Therefore, it is necessary to potentiate programmes that encourage physical exercise or PA and sport for people with a disability where the social, play or health aspects are paramount.

Bearing in mind that previous studies on people without a disability [27] report that there can be differences according to gender in the perception of barriers to the practice of PA and sport and that Gallego et al. [14] state that women with a disability have a lower level of PA and sport practice, it could be interesting to analyse whether the gender of the participants influences the motives and barriers for practising PA and sport in people with a disability. The results obtained in the present study show that women perceive more barriers to the practise of PA and sport than men. These findings coincide with those of Juanbeltz et al. [32] carried out with physically active people with a disability. These differences in the perception of the barriers according to gender can be because women with a disability are in a situation of double discrimination, the barriers due to being a woman and, in turn, the barriers for having a disability, which means that these social and cultural factors place women with a disability in a serious situation of exclusion and discrimination [35,36], increasing the value assigned to the barriers to the practice of PA and sport. With regard to the motives, there are also differences according to gender in various items; while the men gave higher scores to motives related to leisure and overcoming challenges, the women gave greater importance to motives of positive sensations, of seeking a healthy life style, of the doctor’s advice and of recovering from illness or injury. As described, on the one hand, men seem to be more active than women and, thus, have a more positive attitude toward the practice of PA and sport [37]. This can be seen, also, in people with a disability as men, through cultural tradition or because they have a more deeply rooted affinity for PA and sport, have had greater accessibility. This aspect can provoke a more positive attitude towards this activity, and the motives are more geared toward leisure and enjoyment. Moreover, it has been described how the motives of women without a disability are more oriented towards health [38], and this result seems to coincide with those obtained from women with a disability.

It is important to attend to the differences related to barriers and motives according to the type of disability, as the practice of physical and sports activities should be suited to the different impairments and needs of this population. The results of the present study show that the most common barriers for people with PD and ID are intrinsic, including “I can’t make time for myself“, “My health is not good“, “I don’t feel like it“, “I am tired“, while for people with VD, environmental barriers were more important, including, for example, “There are no spaces nearby where I can practise”. This barrier referred to by people with VD leads us to wonder whether there are still facilities and infrastructures that are not accessible to everyone so that people with VD continue to face these environmental barriers [39]. It could be necessary to analyse in the near future whether society in general and, specifically, facilities for the practice of PA and sport have made improvements regarding the accessibility for people with PD and ID but not so much for those with VD and whether this aspect conditions the different perceptions of barriers in these groups. Regarding motives, the participants with PD in the present study highlighted those oriented towards physical improvement and rehabilitation, such as “To control my weight“ or “To recover from an illness or injury”. In contrast, people with ID or VD focussed on motives of leisure, enjoyment and social relations, such as ”To make contact with other people”, “Because I like doing this physical activity” or “For enjoyment”. These results coincide with the findings of the study by Juanbeltz et al. [32]. It would be interesting to know whether the people with PD have a worse condition of physical health than people with other disabilities and whether this could be the reason for the need to gear the practice of PA and sport to the aim of rehabilitation or improving physical health. 

The present study is not without limitations, and despite having used utmost scientific and methodological rigour, as well as a large sample of people with a disability, it should be indicated that the sample used in the study is not representative of the population with a disability, as all the participants are users of programmes of physical exercise or sport and belong to one single country and one single autonomous region. Taking into account that all the subjects are practitioners of physical activity, it may be that the very possibility of self-selection of the activity prevents the subjects from perceiving the barriers related to the environment and, therefore, affects the results of this work. Moreover, their level of practice of PA and sport and the type of activity performed by the participants in the study were not recorded. Similarly, the present study did not analyse the influence of the barriers and motives in the level of practice of PA and sport. Therefore, it would be interesting to carry out similar studies with samples from other countries and with other conditions to be able to observe the possible differences, as well as to measure the levels of practice, the type of PA and sport performed and the health characteristics to see their influence on the motives for and barriers to the practice of PA and sport.

## 5. Conclusions

The results obtained in the present study show that the most important barriers for people with a disability are intrinsic/personal ones. Moreover, the most popular motives were those related to leisure, enjoyment and social aspects. Regarding differences according to gender, it was observed that women perceive more barriers than men. The type of disability can also affect the perceived motives and barriers. These findings should be taken into account by the different professionals and managers of PA and sport to be able to provide an attractive supply of PA to people with a disability. Furthermore, it seems important to bear in mind the barriers and motives when designing different physical exercise and physical-sports activities to be able to increase the practice of PA and sport in people with a disability and, consequently, improve their health.

## Figures and Tables

**Table 1 ijerph-20-01320-t001:** Results (%) referring to the barriers to the practice of physical activity and sport perceived by all the participants in the study.

	Not at All	Somewhat	Quite a Lot	A Lot
I don’t think it is necessary to do exercise	89.2	4.9	4.9	1.0
I don’t like it	85.1	8.9	4.0	2.0
I am too old	82.5	9.7	3.9	3.9
I don’t feel able	81.6	13.6	4.9	0.0
I have to look after my children	80.6	7.8	5.8	5.8
My current job prevents me	78.2	16.8	3.0	2.0
I am embarrassed to be seen doing it	77.7	15.5	2.9	3.9
I don’t know where to do exercise	77.7	9.7	4.9	7.8
I have to look after older relatives	74.8	12.6	7.8	4.9
There is no provision for the activity I like	72.8	14.6	6.8	5.8
It is too expensive	71.8	20.4	6.8	1.0
The spaces where I can practise are not suitable	68.9	19.4	9.7	1.9
Difficulties with transport	67.0	20.4	7.8	4.9
I am tired	66.0	19.4	10.7	3.9
I have no one to do exercise with	65.7	20.6	7.8	5.9
I don’t feel happy with my body	63.0	21.0	13.0	3.0
Fear of hurting myself	61.2	26.2	7.8	4.9
There are no spaces nearby where I can practise	60.8	20.6	11.8	6.9
I am not fit	59.8	24.5	11.8	3.9
I haven’t got the habit	57.3	25.2	11.7	5.8
I don’t feel like it	56.3	20.4	14.6	8.7
I have to do the housework	52.9	23.5	18.6	4.9
The technical staff are not suitable	52.5	17.8	17.8	11.9
My health is not good	51.5	27.2	12.6	8.7
I can’t make time for myself	50.5	33.0	14.6	1.9
Incompatible timetables	47.6	37.9	9.7	4.9
I have a disability	44.1	16.7	17.6	21.6

**Table 2 ijerph-20-01320-t002:** Results (%) referring to the motives for practising physical activity and sport perceived by all the participants in the study.

	Not at All	Somewhat	Quite a Lot	A Lot
To share activities with my children	87.1	8.9	1.0	3.0
To spend more time with my partner	79.2	8.9	6.9	5.0
To recover from an illness or injury	66.0	15.5	7.8	10.7
Out of obligation	62.1	17.5	9.7	10.7
Because of my doctor’s advice	46.6	18.4	25.2	9.7
Because it allows me to be in natural surroundings	33.0	23.3	19.4	24.3
To control my weight	30.1	14.6	30.1	25.2
Because I feel bad if I don’t practise	28.7	35.6	19.8	15.8
Because it’s time just for me	20.6	15.7	27.5	36.3
To improve my appearance	18.4	28.2	30.1	23.3
Because I like to overcome challenges	17.6	11.8	30.4	40.2
To eliminate stress	14.7	35.3	21.6	28.4
Because I enjoy learning new things	12.7	17.6	22.5	47.1
To meet my friends	7.8	23.3	22.3	46.6
To make contact with other people	6.8	14.6	28.2	50.5
Because I want it to be part of my life style	5.9	7.8	42.2	44.1
To keep fit	4.9	16.5	32.0	46.6
To improve my mood	4.9	23.5	39.2	32.4
To improve my mental wellbeing	3.9	23.3	33.0	39.8
To improve my physical capacity	3.9	16.5	38.8	40.8
To disconnect for a while	2.9	19.4	36.9	40.8
Because I like doing this physical activity	2.0	8.8	16.7	72.5
Because I feel good when I do physical activity	1.9	12.6	22.3	63.1
For enjoyment	1.9	11.7	26.2	60.2
To improve my physical health	1.9	18.4	35.0	44.7

**Table 3 ijerph-20-01320-t003:** Results (%) referring to the differences in the perceived barriers to practising physical activity and sport according to gender.

	WOMEN				MEN				Chi^2^
	Not at All	Somewhat	Quite a Lot	A Lot	Not at All	Somewhat	Quite a Lot	A Lot	*p*
Fear of hurting myself	55.0	25.0	17.5	2.5	65.1	27.0	1.6	6.3	0.027
I am embarrassed to be seen doing it	75.0	15.0	5.0	5.0	79.4	15.9	1.6	3.2	NS
I don’t feel happy with my body	60.0	20.0	17.5	2.5	65.0	21.7	10.0	3.3	NS
My current job prevents me	79.5	12.8	5.1	2.6	77.4	19.4	1.6	1.6	NS
I am too old	67.5	15.0	7.5	10.0	92.1	6.3	1.6	0.0	0.007
I am tired	60.0	20.0	15.0	5.0	69.8	19.0	7.9	3.2	NS
It is too expensive	67.5	27.5	5.0	0.0	74.6	15.9	7.9	1.6	NS
The technical staff are not suitable	33.3	30.8	20.5	15.4	64.5	9.7	16.1	9.7	0.010
The spaces where I can practise are not suitable	72.5	17.5	10.0	0.0	66.7	20.6	9.5	3.2	NS
There are no spaces nearby where I can practise	64.1	17.9	10.3	7.7	58.7	22.2	12.7	6.3	NS
I don’t like it	79.5	10.3	10.3	0.0	88.7	8.1	3.2	0.0	0.046
I can’t make time for myself	45.0	35.0	20.0	0.0	54.0	31.7	11.1	3.2	NS
I don’t feel like it.	37.5	32.5	20.0	10.0	68.3	12.7	11.1	7.9	0.016
I don’t know where to do exercise	80.0	10.0	2.5	7.5	76.2	9.5	6.3	7.9	NS
I have to look after my children	70.0	10.0	12.5	7.5	87.3	6.3	1.6	4.8	NS
I don’t feel able	72.5	17.5	10.0	0.0	87.3	11.1	1.6	0.0	NS
I am not fit	42.5	32.5	20.0	5.0	71.0	19.4	6.5	3.2	0.029
Difficulties with transport	67.5	22.5	5.0	5.0	66.7	19.0	9.5	4.8	NS
I have no one to do exercise with	55.0	20.0	17.5	7.5	72.6	21.0	1.6	4.8	0.026
I have to do the housework	40.0	30.0	20.0	10.0	61.3	19.4	17.7	1.6	NS
My health is not good	32.5	32.5	17.5	17.5	63.5	23.8	9.5	3.2	0.007
Incompatible timetables	35.0	52.5	7.5	5.0	55.6	28.6	11.1	4.8	NS
I haven’t got the habit	47.5	27.5	17.5	7.5	63.5	23.8	7.9	4.8	NS
There is no provision for the activity I like	65.0	20.0	12.5	2.5	77.8	11.1	3.2	7.9	NS
I don’t think it is necessary to do exercise	84.6	7.7	7.7	0.0	92.1	3.2	3.2	1.6	NS
I have to look after older relatives	67.5	17.5	10.0	5.0	79.4	9.5	6.3	4.8	NS
I have a disability	32.5	17.5	27.5	22.5	51.6	16.1	11.3	21.0	NS

NS = non-significant differences.

**Table 4 ijerph-20-01320-t004:** Results (%) referring to the differences in the perceived motives for practising physical activity and sport according to gender.

	WOMEN				MEN				Chi^2^
	Not at All	Somewhat	Quite a Lot	A Lot	Not at All	Somewhat	Quite a Lot	A Lot	*p*
To recover from an illness or injury	50.0	20.0	15.0	15.0	76.2	12.7	3.2	7.9	0.030
To control my weight	20.0	10.0	37.5	32.5	36.5	17.5	25.4	20.6	NS
To keep fit	2.5	12.5	40.0	45.0	6.3	19.0	27.0	47.6	NS
Because I feel good when I do physical activity	2.5	15.0	30.0	52.5	1.6	11.1	17.5	69.8	NS
Because it’s time just for me	25.0	20.0	32.5	22.5	17.7	12.9	24.2	45.2	NS
To meet my friends	2.5	27.5	25.0	45.0	11.1	20.6	20.6	47.6	NS
For enjoyment	2.5	20.0	35.0	42.5	1.6	6.3	20.6	71.4	0.025
Because I enjoy learning new things	15.4	25.6	20.5	38.5	11.1	12.7	23.8	52.4	NS
To improve my physical health	2.5	15.0	37.5	45.0	1.6	20.6	33.3	44.4	NS
To improve my mental wellbeing	5.0	25.0	42.5	27.5	3.2	22.2	27.0	47.6	NS
Because I like doing this physical activity	2.5	7.5	22.5	67.5	1.6	9.7	12.9	75.8	NS
To eliminate stress	12.5	45.0	30.0	12.5	16.1	29.0	16.1	38.7	0.018
To improve my mood	5.1	23.1	51.3	20.5	4.8	23.8	31.7	39.7	NS
To make contact with other people	5.0	12.5	37.5	45.0	7.9	15.9	22.2	54.0	NS
To spend more time with my partner	66.7	20.5	7.7	5.1	87.1	1.6	6.5	4.8	0.012
Because I feel bad if I don’t practise	23.1	38.5	25.6	12.8	32.3	33.9	16.1	17.7	NS
Because I like to overcome challenges	23.1	15.4	43.6	17.9	14.3	9.5	22.2	54.0	0.004
Because I want it to be part of my life style	0.0	10.0	57.5	32.5	9.7	6.5	32.3	51.6	0.019
For the positive sensations that doing physical activity gives me	0.0	5.0	42.5	52.5	3.2	9.5	17.5	69.8	0.033
Because it allows me to be in natural surroundings	45.0	27.5	15.0	12.5	25.4	20.6	22.2	31.7	NS
Out of obligation	52.5	25.0	12.5	10.0	68.3	12.7	7.9	11.1	NS
Because of my doctor’s advice	30.0	15.0	37.5	17.5	57.1	20.6	17.5	4.8	0.007
To improve my appearance	12.5	20.0	45.0	22.5	22.2	33.3	20.6	23.8	NS
To improve my physical capacity	5.0	15.0	50.0	30.0	3.2	17.5	31.7	47.6	NS
To disconnect for a while	5.0	27.5	45.0	22.5	1.6	14.3	31.7	52.4	0.021
To share activities with my children	78.9	15.8	2.6	2.6	92.1	4.8	0.0	3.2	NS

NS = non-significant differences.

**Table 5 ijerph-20-01320-t005:** Results (%) referring to the differences in the perceived barriers to practising PA and sport according to the type of disability.

	PD				ID				VD				Chi^2^
	Not at All	Somewhat	Quite a Lot	A Lot	Not at All	Somewhat	Quite a Lot	A Lot	Not at All	Somewhat	Quite a Lot	A Lot	*p*
Fear of hurting myself	70.0	20.0	10.0	0.0	48.6	34.3	8.6	8.6	62.9	25.7	5.7	5.7	NS
I am embarrassed to be seen doing it	60.0	26.7	3.3	10.0	82.9	14.3	2.9	0.0	85.7	8.6	2.9	2.9	NS
I don’t feel happy with my body	50.0	26.7	23.3	0.0	65.6	9.4	15.6	9.4	74.3	22.9	2.9	0.0	0.020
My current job prevents me	83.3	10.0	3.3	3.3	87.9	12.1	0.0	0.0	65.7	25.7	5.7	2.9	NS
I am too old	73.3	13.3	6.7	6.7	82.9	8.6	5.7	2.9	91.4	8.6	0.0	0.0	NS
I am tired	66.7	26.7	3.3	3.3	60.0	14.3	20.0	5.7	77.1	20.0	2.9	0.0	0.001
It is too expensive	60.0	26.7	10.0	3.3	71.4	25.7	2.9	0.0	85.7	5.7	8.6	0.0	NS
The technical staff are not suitable	31.0	27.6	27.6	13.8	68.6	14.3	5.7	11.4	55.9	11.8	20.6	11.8	NS
The spaces where I can practise are not suitable	76.7	20.0	3.3	0.0	74.3	20.0	5.7	0.0	57.1	20.0	17.1	5.7	NS
There are no spaces nearby where I can practise	66.7	23.3	3.3	6.7	77.1	11.4	11.4	0.0	41.2	29.4	14.7	14.7	0.006
I don’t like it	86.2	10.3	3.4	0.0	70.6	14.7	8.8	5.9	97.1	2.9	0.0	0.0	NS
I can’t make time for myself	30.0	30.0	40.0	0.0	71.4	20.0	2.9	5.7	48.6	45.7	5.7	0.0	0.000
I don’t feel like it	50.0	26.7	16.7	6.7	45.7	17.1	17.1	20.0	77.1	14.3	8.6	0.0	0.017
I don’t know where to do exercise	76.7	16.7	3.3	3.3	77.1	5.7	5.7	11.4	77.1	8.6	5.7	8.6	NS
I have to look after my children	66.7	13.3	10	10.0	94.3	5.7	0.0	0.0	82.9	2.9	8.6	5.7	0.038
I don’t feel able	80.0	13.3	6.7	0.0	77.1	17.1	5.7	0.0	88.6	8.6	2.9	0.0	NS
I am not fit	60.0	23.3	6.7	10.0	61.8	14.7	20.6	2.9	60.0	31.4	8.6	0.0	NS
Difficulties with transport	66.7	26.7	6.7	0.0	65.7	25.7	5.7	2.9	65.7	11.4	11.4	11.4	NS
I have no one to do exercise with	56.7	20.0	13.3	10.0	61.8	23.5	5.9	8.8	74.3	20.0	5.7	0.0	NS
I have to do the housework	43.3	26.7	23.3	6.7	61.8	17.6	14.7	5.9	57.1	25.7	14.3	2.9	NS
My health is not good	33.3	30.0	20.0	16.7	51.4	40.0	5.7	2.9	68.6	14.3	11.4	5.7	0.026
Incompatible timetables	33.3	50.0	10.0	6.7	62.9	31.4	2.9	2.9	45.7	37.1	14.3	2.9	NS
I haven’t got the habit	36.7	30.0	23.3	10.0	62.9	20.0	8.6	8.6	71.4	22.9	5.7	0.0	NS
There is no provision for the activity I like	66.7	16.7	6.7	10.0	77.1	8.6	5.7	8.6	77.1	17.1	5.7	0.0	NS
I don’t think it is necessary to do exercise	83.3	13.3	3.3	0.0	82.9	2.9	11.4	2.9	100.0	0.0	0.0	0.0	NS
I have to look after older relatives	76.7	16.7	6.7	0.0	71.4	11.4	8.6	8.6	82.9	5.7	8.6	2.9	0.022
I have a disability	27.6	20.7	20.7	31.0	48.6	14.3	22.9	14.3	54.3	17.1	8.6	20.0	NS

PD = physical disability, ID = intellectual disability, VD = visual disability, NS = non-significant differences.

**Table 6 ijerph-20-01320-t006:** Results (%) referring to the differences in the perceived motives for practising PA and sport according to type of disability.

	PD				ID				VD				Chi^2^
	Not at All	Somewhat	Quite a Lot	A lot	Not at All	Somewhat	Quite a Lot	A Lot	Not at All	Somewhat	Quite a Lot	A Lot	*p*
To recover from an illness or injury	36.7	20.0	13.3	0.0	71.4	17.1	5.7	5.7	88.6	5.7	5.7	0.0	0.000
To control my weight	16.7	13.3	36.7	33.3	17.1	20.0	28.6	34.3	57.1	11.4	22.9	8.6	0.007
To keep fit	3.3	10.0	26.7	60.0	5.7	14.3	31.4	48.6	5.7	25.7	31.4	37.1	NS
Because I feel good when I do physical activity	0.0	10.0	36.7	53.3	5.7	20.0	11.4	62.9	0.0	5.7	20.0	74.3	NS
Because it’s time just for me	20.7	17.2	37.9	24.1	20.0	8.6	20.0	51.4	20.0	22.9	25.7	31.4	NS
To meet my friends	13.3	30.0	26.7	30.0	8.6	11.4	14.3	65.7	2.9	25.7	28.6	42.9	NS
For enjoyment	0.0	20.0	33.3	46.7	5.7	14.3	22.9	57.1	0.0	0.0	20.0	80.0	0.022
Because I enjoy learning new things	20.0	20.0	26.7	33.3	14.3	25.7	14.3	45.7	2.9	8.6	25.7	62.9	NS
To improve my physical health	3.3	13.3	40.0	43.3	2.9	25.7	20.0	51.4	0.0	17.1	42.9	40.0	NS
To improve my mental wellbeing	3.3	43.3	26.7	26.7	8.6	20.0	25.7	45.7	0.0	8.6	42.9	48.6	0.028
Because I like doing this physical activity	0.0	6.7	23.3	70.0	5.9	11.8	17.6	64.7	0.0	2.9	11.4	85.7	0.013
To eliminate stress	10	56.7	16.7	16.7	26.5	23.5	23.5	26.5	5.7	31.4	20.0	42.9	0.013
To improve my mood	10.3	27.6	41.4	20.7	5.7	20.0	31.4	42.9	0.0	20.0	45.7	34.3	NS
To make contact with other people	13.3	20.0	40.0	26.7	8.6	2.9	20.0	68.6	0.0	17.1	28.6	54.3	0.005
To spend more time with my partner	86.7	6.7	3.3	3.3	70.6	11.8	8.8	8.8	80.0	8.6	8.6	2.9	NS
Because I feel bad if I don’t practise	30.0	40.0	26.7	3.3	42.9	34.3	11.4	11.4	11.8	32.4	23.5	32.4	0.028
Because I like to overcome challenges	23.3	16.7	36.7	23.3	28.6	11.4	14.3	45.7	0.0	8.6	40.0	51.4	0.016
Because I want it to be part of my life style	3.3	3.3	56.7	36.7	14.3	8.6	31.4	45.7	0.0	5.9	44.1	50.0	0.003
For the positive sensations that doing physical activity gives me	0.0	6.7	40.0	53.3	5.7	17.1	22.9	54.3	0.0	0.0	20.0	80.0	NS
Because it allows me to be in natural surroundings	30.0	30.0	26.7	13.3	40.0	5.7	17.1	37.1	25.7	37.1	14.3	22.9	0.044
Out of obligation	70.0	23.3	6.7	0.0	60.0	17.1	11.4	11.4	57.1	11.4	11.4	20.0	NS
Because of my doctor’s advice	23.3	26.7	33.3	16.7	42.9	17.1	31.4	8.6	74.3	14.3	8.6	2.9	0.003
To improve my appearance	13.3	26.7	43.3	16.7	11.4	25.7	25.7	37.1	31.4	34.3	22.9	11.4	0.047
To improve my physical capacity	3.3	13.3	53.3	30.0	8.6	11.4	28.6	51.4	0.0	22.9	40.0	37.1	NS
To disconnect for a while	6.7	33.3	36.7	23.3	2.9	17.1	37.1	42.9	0.0	8.6	34.3	57.1	NS
To share activities with my children	79.3	13.8	0.0	6.9	97.1	2.9	0.0	0.0	88.2	8.8	0.0	2.9	0.000

PD = physical disability, ID = intellectual disability, VD = visual disability, NS = non-significant differences.

## Data Availability

Data collection is not available for privacy reasons.

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
