# Peer review of "Analysis of the Barriers and Motives for Practicing Physical Activity and Sport for People with a Disability: Differences According to Gender and Type of Disability"

_ijerph, 2023, doi:10.3390/ijerph20021320_

Round 1

Reviewer 1 Report

Dear authors and respected colleagues,

it was a pleasure to read your work. The topic of the work is very interesting, but unjustifiably neglected in society. I believe that your work in the future will be the basis for further research on this topic and will help people with disabilities to integrate into society in which they will be more physically active and thus improve their psychophysical health and quality of life.

Good work.

Reviewer 2 Report

Nicely written.  Please see comments in attachment.

Reviewer 3 Report

The introduction is well planned as well as the objectives.

However, hypotheses are not included. These should be reflected in the paper

It would be interesting to observe the differences in motives and barriers depending on the type of disability. Would it be possible to present these data?

The methodology does not provide information about the reliability indices of the instrument used. It is recommended that it be included.

Include cronbach's alpha obtained for this sample

Round 2

Reviewer 3 Report

The authors have incorporated the requested suggestions into the paper